# Analysis of the Effects of Lockdown on Staff and Students at Universities in Spain and Colombia Using Natural Language Processing Techniques

**DOI:** 10.3390/ijerph19095705

**Published:** 2022-05-07

**Authors:** Mario Jojoa, Begonya Garcia-Zapirain, Marino J. Gonzalez, Bernardo Perez-Villa, Elena Urizar, Sara Ponce, Maria Fernanda Tobar-Blandon

**Affiliations:** 1Department of Computer Science, Engineering Faculty, Electronics and Telecommunications University of Deusto, 48014 Bilbao, Spain; mbgarciazapi@deusto.es; 2Unit of Public Policy, Simon Bolivar University, Caracas 89000, Venezuela; margonza@usb.ve; 3Heart, Vascular and Thoracic Institute, Cleveland Clinic Florida, Weston, FL 33331, USA; perezbernardoandres@gmail.com; 4Deusto Business School Health, University of Deusto, 48014 Bilbao, Spain; elena.urizar@deusto.es; 5International Research Projects Office (IRPO), University of Deusto, 48014 Bilbao, Spain; sponce@deusto.es; 6Public Health School, Universidad del Valle, Cali 76001, Colombia; maria.f.tobar@correounivalle.edu.co

**Keywords:** COVID-19, university student, socio-demographic factors, satisfaction, perception, online learning, mental health, habits, institutions, continents, natural language processing, Swivel embedding, word cloud

## Abstract

The aim of this study is to analyze the effects of lockdown using natural language processing techniques, particularly sentiment analysis methods applied at large scale. Further, our work searches to analyze the impact of COVID-19 on the university community, jointly on staff and students, and with a multi-country perspective. The main findings of this work show that the most often related words were “family”, “anxiety”, “house”, and “life”. Besides this finding, we also have shown that staff have a slightly less negative perception of the consequences of COVID-19 in their daily life. We have used artificial intelligence models such as swivel embedding and a multilayer perceptron as classification algorithms. The performance that was reached in terms of accuracy metrics was 88.8% and 88.5% for students and staff, respectively. The main conclusion of our study is that higher education institutions and policymakers around the world may benefit from these findings while formulating policy recommendations and strategies to support students during this and any future pandemics.

## 1. Introduction

Since the identification of the severe acute respiratory novel coronavirus 2 (SARS-CoV-2) in December of 2019 in Wuhan, China, it has provoked turbulence in our society and countries worldwide. Thereby, rigorous public health measures were imposed around the world, such as quarantine, testing, and social distancing. However, they were different around the world. In Latin America, the different governments imposed confinement and social distancing measures in different ways, in some cases based on the reality of the moment and others on what was happening in countries outside the continent. Thus, premature and unplanned strategies unleashed a series of effects that added complexity to the pandemic situation, mostly related to inequity in health and the economy [1]. In Europe, the effects of the pandemic spread in a much more accelerated manner due to its high international traffic by air, land, and sea. This situation determined that the governments quickly assumed confinement measures, which led to important effects on economic, political, and social aspects of life.

The COVID-19 pandemic created unexpected challenges that affected each one of us, having disruptive effects and a significant impact on almost all of the sectors of our society around the world: health [2,3], the economy [4], and education [5] were not exceptions. In fact, the academic community is a population group that has experienced dramatic effects during the pandemic, especially in the first five months of 2020, impacting their daily lives and the prospects for their immediate and distant futures.

According to UNESCO’s monitoring, more than 160 countries implemented nationwide closures, which impacted over 87% of the world’s student population [6]. Physically closing or restricting access to educational institutions combined with other policy measures that contributed to reducing the spread of the COVID-19 pandemic. However, the pandemic has posed challenges for the academic community and their families, friends, employers, and the global economy [5]. The closure of educational institutions on all levels has provoked learners to stay at home and this situation reached a peak of 1.598 billion students at home from 194 countries on 1 April 2020 [1]. This situation led to the alteration of the patterns of teaching, the transition to online learning, the change of the communication channels between academic staff members and the teachers and students, new assessment methods, different workloads, different performance levels [5], the use of new technology, and other outcomes. All of these measures gave way to a global change that was already underway, leaving a gap of inequality and poverty in the world’s poorest regions. This inequality and poverty was accelerated with the pandemic and, therefore, this widened the gap of social injustice, poverty, and social inequity.

The investment of resources to strengthen the technology in those regions and countries requiring it is a priority. There should be a significant investment in these countries concerning the enhancement of connectivity networks in distant regions and those with high geographical dispersion. Apart from the use of the internet and technology, there is still a significant percentage of illiteracy in most Central and South American countries which hampers the adequate use thereof, which must be overcome in order to reduce inequity in education [7].

Under this scenario, there is a need to collect more expansive data in order to understand how COVID-19 reshapes academic life in higher education. Although several papers have already been published that analyze the various aspects of the COVID-19 pandemic on academic life and the consequences for physical and mental health, the economy, society, and the environment [5], most of them focus on the students, disregarding staff, and are limited to one country or a single higher education institution. Additionally, it has been observed in the literature review that previous works have not used a sentiment analysis technique at a large scale for the analysis of their data. The idea to develop a natural language processing model to perform an automatic classification of the text in sentiment categories that are related to what people felt in lockdown may allow community leaders to perform earlier decision making. Furthermore, the generation of strategies that are based on a local environment is possible with this kind of tool.

According to the source of information that was used, sentiment analysis can be divided into two groups: social network refereed papers [8], especially those involving Twitter [9], and texts from interviews with family members [10]. In the COVID-19 pandemic, the most frequent source of information for conducting sentiment analysis has been the use of Twitter and other social network content to categorize motivations that are related to multiple infectious diseases and in particular relation to the monitoring, data analysis, and challenges that are faced by researchers in terms of available information and to the relationship between social media platforms and the community [11]. The use of sentiment analysis through interviews with the family members of COVID-19 patients who used the virtual modality of visits focused on the feelings that they experienced at the visit, the barriers, their concerns about this option, and opportunities that they noted for improving the method [12].

However, there have been no reports of conducting sentiment analysis using expressions in online public opinion surveys to date. The aim of our study is to understand the students’ and staff’s perspectives and experiences of how the pandemic and the closure of universities have had an impact on the university community, students, and staff in Spain and Colombia by applying sentiment analysis methods. This paper attempts to shed light on the impact of the COVID-19 pandemic on staff and students from universities in Spanish-speaking countries. This article focuses on identifying what is the perception of the students and staff of the universities in Colombia and Spain, regarding the contingent measures that were taken by the governments of the different countries that integrated the strategies of lockdown and social distance. For this aim, we conducted a customized questionnaire in order to understand how students perceive the impacts of the first wave of the COVID-19 crisis in early 2020 on various aspects of their lives on a global level and we applied sentiment analysis techniques.

## 2. Materials and Methods

### 2.1. Online Survey

The online questionnaire is part of a global initiative, The Lockdown Project, which was led by London School of Economics (LSE). The objective of this project was to understand the experiences of university staff and students around the world and to construct a better factual map of what was happening to them under the COVID-19 situation.

The survey has been distributed in more than 25 countries by LSE and in collaboration with the project partners, 18 collaborators, and 11 supporting organizations. The survey was initially designed in English and translated into 17 languages. This study is based on the Spanish version of the survey.

The data were obtained through a web-based comprehensive questionnaire that was composed of 68 questions. Of these, 67 are closed-ended questions that cover different aspects: socio-demographic, geographic, and the impact of the COVID-19 pandemic in their daily lives (studies, work, families, social life, habits, etc.). The last question is an open-end question that is related to personal reflections: “Share any other thoughts/experiences about your life in lockdown” and set the basis for the analysis of this paper.

The scope of the present study has limitations due to various causes, among them the little information that was provided by the governments of some countries, in several cases hasty decisions or those which were unanalyzed, and the media’s impact that generated great uncertainty, lack of planning, and false news leading to conspiracy theories that ended up affecting the perception of young students about confinement and the true impact of the pandemic in their lives. It is necessary, as already said before, that more studies of this type be carried out in order to recognize the perceptions and affirm or deny what was initially given as the perception of the study participants, since this is a qualitative study.

### 2.2. Study Participants and Procedure

All of the participants consented to their participation and responded to the survey in an anonymized way in order to ensure that no personal data were collected. All of the data that were collected were GDPR-compliant and have been used for research purposes only.

The target population comprised university staff and students of Spanish-speaking countries who were at least 18 years old. The web-based survey was launched via the open-source web (https://research.healthbit.com/c/LockedDown-en accessed on 28 May 2020) on 20 April and it remained open until July. The survey was distributed by advertising on the University of Deusto communication systems and social media, dissemination through two university networks (Unijes and AUSJAL), and targeted mailing to other Spanish and Latin America universities. We obtained responses from 2 countries. The distribution by country and respondent’s profile (student or staff member) is depicted in Table 1 and Figure 1.

### 2.3. Labeling

The open responses to the question were labeled as positive, negative, or neutral by two independent researchers for the sentiment analysis. Their responses and doubts were resolved by a third independent researcher.

The criteria that we applied were the following:If the sentences indicated a positive testimony the associated label is positive, i.e., “There are some aspects of home-schooling which I like and value” or “The pandemics have shown us to live better with less money”.On the other extreme, if the testimony was clearly negative, the associated label was negative, i.e., “I was not able to progress with my research thesis and I felt anxious”, or “The social relationships are not the same in remote areas, my friendship relationships have deteriorated”.The neutral testimonies have received a neutral tag: “In the end, we will all have COVID-19, we will normalize it, and we’ll learn to live with it”, “It’s time to reflect”.

If a single testimony had inputs corresponding to more than one of our three labeling categories, we counted each input under the corresponding labeling item. The size of our hand-labeled data allowed us to perform cross validation experiments and check for the variance in the performance of the classifier across folds.

#### 2.3.1. Histograms of Labeled Text by Sentiment Categories

The distribution of the data by categories gave us a clear idea of the tendency of the feelings in the categories proposed for each country that was selected as a case study. For this reason, the distribution charts of the corresponding data for the student and staff groups have been prepared separately.

##### Histograms for Student Group Data

The following histogram charts correspond to the data distributions by country and by sentiment category for the student data set.

In the case of Spain, we can observe that most of the answers have a negative sentiment compared to the prevalence of the positive and neutral categories. The numbers of labels per class for this country are: 88 negative, 6 neutral, and 12 positive. In the case of Colombia, we can observe that most of the answers have a negative sentiment compared to the prevalence of the positive and neutral categories. The number of labels per class for this country are: 84 negative, 26 neutral and 9 positive.

##### Histograms for the Staff Dataset

The Figure 2 shows the histogram charts correspond to the data distribution by country and by sentiment category for the staff data set.

In the case of Spain, we can observe that most of the answers have a negative sentiment, compared to the prevalence of the positive and neutral categories. The numbers of labels per class for this country are: 41 negative, 14 neutral and 27 positive. In the case of Colombia, we can observe most of the answers have a negative sentiment compared to the prevalence of the positive and neutral categories. The number of labels per class for this country are: 30 negative, 14 neutral and 14 positive. We showed this in Figure 3.

In general, we can observe for the students in both countries that the corresponding texts are mostly negative in their sentiment with 185 labels of this category, in contrast to 35 labels for the neutral category and 23 for the positive category. For the staff group, the corresponding texts are mostly negative in their sentiment with 71 labels of this class, in contrast with 28 labels for the neutral category and 41 for the positive category.

### 2.4. NLP Techniques

Natural language processing techniques let us automate the analysis of the content of a text with the aim of extracting knowledge. In the context of this research, we are mainly interested in sentiment analysis, as well as in the infographic presentation of the text that was written by the users who answered the survey. Therefore, we present two subsections with the techniques that were used to achieve this goal. The general proposed solution is presented in Figure 4.

In the next subsection, we are going to describe in detail the stages of the proposed solution, which is represented in the diagram Figure 5.

#### 2.4.1. Sentiment Analysis

In order to analyze the sentiment of a text, it is necessary to represent it numerically; to do this we have used the “word embedding” technique, which is explained in Section 2.4.1.1. This method encodes the words into numerical vectors without losing the contextual information. It is important to note that this technique requires a large amount of data in the training stage. Therefore, we decided to use transfer learning from the IMDB database, which is appropriate [13] for this type of task and the details of which are explained in Section 2.4.1.2. Once the task of encoding the data into a numeric domain was performed, a classifier was trained to determine the sentiment of the text, figuring out the category, in this case positive, negative, or neutral. Details of the classification methods are given in Section 2.4.1.2.

##### 2.4.1.1. Word Embedding

Word embedding is a natural language processing (NLP) approach that is used to convert words into vector arrays, with the intention of capturing the semantic and syntactic relationship between words, with the purpose of simulating the human learning of linguistic vocabulary. This problem is one of the most interesting of this field, per [14] in which the authors say, “Representation learning is a long-standing problem in natural language processing (NLP)”. In order to solve this representation problem, we have decided to go beyond the surface forms of a text (e.g., symbols, words, sentences, and actual documents) to the meaningful similarities (e.g., semantic or syntactic) between two text fragments [15].

The main idea is to represent each word in a large body of text by a feature vector, so that it is possible to measure the similarity between vectors (i.e., words) using linear algebra (e.g., using cosine similarity [16]). There are two categories of word embedding models, on the one hand is the models that are based on matrix factorization and on the other hand the models that are based on sampling from a sliding window [17]. This approach has been shown to be of great utility in tasks such as translation, analysis, and the ascertainment of word similarity [18]. Currently, there are several successful and well-known word embedding models, such as GloVe [19] and Word2Vec [20], which have had a profound impact on NLP research and have inspired the construction of new word vectors based on a stochastic downward gradient.

##### 2.4.1.2. Swivel Embedding

Swivel [21] (Submatrix-wise Vector Embedding Learner) is a model that proposes a hybrid between the shortcomings of the SkipGram Negative Sampling (SGNS) model [22] and GloVe [19]. On the one hand, it uses a co-occurrence matrix to calculate the PMI (point-wise mutual information) between pairs of words, which it uses as an optimization objective by decreasing the error, using stochastic gradient descent, between the dot product of the weight vectors (embeddings) of core words and context words and the PMI as calculated through the statistical counting of the co-occurrence of words within the corpus. One of the outstanding advantages is the possibility of performing distributed training, since the nature of the proposal is to divide the co-occurrence matrix into k sub-matrices, extending the capability to parallelized training in a workers and central server configuration. This in turn allows the training of word embeddings with larger corpora, adding that the computational cost is proportional to the size of the co-occurrence matrix, as opposed to SGNS, the computational cost of which is proportional to the size of the corpus. One of the advantages compared to GloVe is that it considers the weighting of unobserved co-occurrences, thus providing a better vector representation for rare words. The model outperforms previous models [23,24] in different NLP tasks such as WordSim similarity and semantic analysis.

##### 2.4.1.3. Transfer Learning with English Google News and the IMDB Database

As mentioned above, a large amount of text is necessary to train stochastic gradient-based models so as to obtain the desired knowledge extraction. The syntactic and semantic relations of the context were obtained using a pre-trained embedding with the English Google News dataset [25], which consists of 130 GB of corpus. It provides a vector encoding of 20 features or dimensions.

On the other hand, the raters were trained with the IMDB database, which consists of 50,000 movie reviews, given by different users from all over the world, on the platform with the same name; separated into 80% for training and 20% for validation. Each comment is labeled with a value of 0 and 1 to discriminate between negative or positive comments accordingly. It is important to highlight that the IMDB database [13] was used as an input to train and validate the model. However, the entire testing stage was performed using the database of open-ended responses that were given in the survey regarding people’s thoughts on quarantine due to COVID-19 disease.

##### 2.4.1.4. Classification Methods: Multilayer Perceptron MLP

The proposed configuration that was required to build a system that is capable of analyzing the sentiment in a text fragment consists of a 20-dimensional word embedding Swivel [26] that is used to find the vector representation of the input text, then a neural network consisting of an input layer of 20 neurons and a hidden layer of 16 neurons, with a Relu activation layer and an output layer of a single neuron with a sigmoidal tangent activation function. The schematic of the proposal can be seen in Figure 6.

The sigmoidal tangent activation function at the output was chosen in order to obtain a continuous value representing the degree of positive sentiment in a text fragment, i.e., a regressor between −1 and 1 was constructed from the dichotomous features 0 and 1, which could usually be used for a classification task.

##### 2.4.1.5. Classification Methods: SVM Support Vector Machine

As in the previous section, the IMDB database was used to train and validate a classifier, the open response database for testing, and an embedding Swivel for text encoding; however, at this point the classifier was based on a support vector machine in order to carry out the regression task. As in the previous model, the output corresponded to a value between −1 and 1 in order to determine the degree of negativity or positivity of the text that was used in the input. Figure 7 shows the proposed model.

##### 2.4.1.6. Decision Model Based on Interval Comparison

In the output of the proposed classification models, an interval-based decision stage was included, the limits of which were obtained through 5-fold cross-validation, since the O values between interval −1 and 1 of the proposed model output must be classified into the positive, neutral or negative classes. It should be noted that the limits for the negative and positive classes were not explicitly found, since they were discarded with the two thresholds that were found. In addition, it should be noted that the positive upper threshold is always 1 and the negative lower threshold is always −1. To achieve this objective, the algorithm that is shown in Figure 8 was applied.

##### 2.4.1.7. Performance Measurement

Finally, the performance of the proposed models was measured, based on the experts’ assessment of the test data; i.e., all of the open responses to the instrument that was used for data collection. With the predictions obtained, the respective confusion matrices were constructed and the accuracy metric was calculated as a representative value in order to compare the performance.

#### 2.4.2. Cloud Words

A word cloud is a graphical representation of the relative frequencies of words in a text, i.e., the number of times a word is repeated within a text [27]. In Figure 9, we present the block diagram that was applied in order to obtain the cloud words. The result was used to analyze the text for the categories negative, neutral and positive.

It is very important to highlight that stop words were eliminated. The threshold frequency that allows the appearance or not of a word within the configured infographic or word cloud is at least 20 repetitions. Finally, the image that is obtained from the use of this process will present the word in a font size that is directly proportional to the number of times that the word was repeated within a document or set of texts.

## 3. Results

This article presents a novel data set of the perceptions and behaviors of university staff and students that were collected after the beginning of the COVID-19 outbreak in two Spanish-speaking countries. In this study we present the summary of the findings from a sample of 225 students and 140 staff from two countries. These data are part of a larger international survey, where students and staff were asked questions about the same topic. The questionnaire was composed of 53 multiple choice questions and one open text question or item.

Our analysis can be used to uncover the impacts of the COVID-19 pandemic on the perceptions and behaviors (work/study, movement, and travel) of Spanish and Latin American university staff and students and to provide further insights into the sensitivity of students and staff to confinement and online learning/teaching.

### 3.1. NLP Techniques

In order to obtain the results that are presented below, 225 records were used in the case of the students and 140 records in the case of the staff, corresponding to the answers to the open question of the instrument that was used for the data collection in this research. Therefore, four confusion matrices were obtained corresponding to the two sentiment analysis models that were described in Section 2.4.1.1 and an image of the infographic that is described in Section 2.4.2 was created.

#### 3.1.1. Sentiment Analysis

Once the proposed classification models were trained and validated, the following results were obtained in the test stage during the task of the classification of the feelings of the input text, which correspond to the opinions that were given by the people who responded to the survey instrument for the collection of the information for this research.

##### Decision Thresholds for Each Case

Table 2 presents the average decision thresholds for each proposed model and the respective dataset as were obtained after the 5-fold cross-validation was applied. It was observed that the range of neutrals for the staff occupies a higher range of values (from 0.38 to 0.52) while for the students it ranges from 0.22 to 0.49.

Table 3 presents the metrics that were achieved for each dataset. The presented values correspond to the accumulate weighted metrics that were obtained for the different reference classes by using the one vs all approach. In this way, the values were obtained for the negative, neutral and positive categories with their respective weights. They were calculated based on the amount of data per each class. As we can observe, for all cases, the MLP-based classifier performed better in terms of text classification. On the other hand, the weighted method let us reduce the bias that was caused by the unbalanced dataset that was used in the present study. The model with the best performance is the same for both the student and staff datasets.

##### Confusion Matrices for the MLP-Based Classifier Model

With the thresholds obtained, we then proceeded to execute the decision stage at the output of the MLP model in order to determine the confusion matrix of Table 4 for the student case.

As is detailed in the following tables, we performed an analysis of the relevant misclassifications results that were obtained for the NLP system (Tables in Appendix A
Table A1, Table A2, Table A3 and Table A4 are in original Spanish language). This analysis was undertaken with the purpose of identifying the text classification’s weaknesses and limitations. As can be seen in the Table 5, one of the inputs by which the system got confused is the word “solidarity”, which by itself does not express any feeling. For the case of the second input of the previous table, this misclassification was due to a mistake in the writing, which changed the idea that the person wanted to express.

The second sentence presented in the Table 6, which the system misclassified, was analyzed resulting in the finding that, in isolation, it can be understood with a positive sentiment; however, it has been labeled as negative as the experts have contextualized the sentence in the framework of a complaint.

The remaining misclassified sentences in the table above are a consequence of the bias of the database, as it should be noted that transfer learning was applied in order to obtain the sentiment opinions in this application, due to the limited number of texts for the training stage.

With the thresholds that were obtained, we proceeded to execute the decision stage at the output of the MLP model, in order to determine the confusion matrix in Table 7 for the staff case.

To produce the following tables, we performed an analysis of the relevant misclassifications results that were obtained for the NLP system again. This analysis was undertaken with the purpose of identifying the weaknesses and limitations of the text classification for the staff dataset. As can be seen in Table 8, the system presents errors when the input sentence is long in its word length and when grammatical and lexical errors are present. That is why the previous examples were classified as negative, when they were labeled as positive by the independent researchers. It should be noted that the system is sensitive to the correct wording of the answers that were given by the users.

The sentence presented in Table 9 is a consequence of the bias of the database, the same as the classification errors that were presented in the data from the group of students, where it was observed that, although it expresses a negative feeling, the natural language processing system catalogs it as positive.

##### Confusion matrices for the SVM-Based Classifier Model

As for the previous model, we proceeded to run the decision stage at the output of the SVM model in order to determine the confusion matrix in Table 10 for the student case.

Finally, we proceeded to execute the decision stage at the output of the SVM model in order to determine the confusion matrix in Table 11 for the staff case.

#### 3.1.2. Cloud Words

The Figure 10 was obtained based on the frequency of the occurrence of each word. In the analyzed text, the word “confinement” is the word which appears the most; however, the words “university” and “pandemic” were observed as following “confinement” in terms of their relative frequency. In addition, the words “anxiety”, “classes”, “people”, and “family”, among others, stand out.

## 4. Discussion

### 4.1. Main Limitations

The main limitations that were encountered, from the point of view of the technology that was used for data analysis, are detailed below:The NLP classification model uses an automatic translator to the English language as the original text is in Spanish, and as a result of this errors could occur during the automatic translation of the texts. In the future, a database could be used to apply transfer learning in the Spanish language, in the stage of the codification of the words (embedding) and in the stage of the classification of the texts.There is sensitivity of the system to errors in spelling and grammar in the texts written by the users, making it difficult for the NLP model to automatically translate and interpret the feeling of the texts.The system is biased by the use of a general-purpose database, such as IMDB, in the application of the transfer learning technique [28], due to the limited number of texts that are used for training from scratch. In the future, researchers may aim to extend the questionnaire to more countries and more users so as to increase the number of texts and thus be able to train a specific model for this application.

First, the use of an automatic Spanish–English translator by the NLP classification model may lead to considerable mistranslations. Further studies may implement in the codification and classification stages a database with transfer learning techniques in Spanish. Besides, the automatic translation could produce a bias since the original expressions could be altered in the translation process. However, an alternative solution is the use of pretrained natural language processing models, which are available in more-common languages. This potential bias was shown to have been overcome our work, since we obtained an acceptable performance compared with the experts’ opinions. In future works, it will be interesting to compare the performance of our work with that of a specific NLP model that is trained with a Spanish corpus for a sentiment text classification task. Besides these considerations, the cultural expressions are also an issue to be considered because different expressions can have different meaning in the same language.

Second, the system’s sensitivity to spelling and grammatical mistakes create difficulty for the automatic translation and therefore the sentiment analysis by the NLP model. Third, the use of the general IMDB database for the transfer learning technique biased the results due to the limited number of texts that were available for data training. Future surveys should include more countries and users in order to improve the data training. Finally, the selection bias due to the use of an online survey limited the sample to those with internet access and reduced our study’s external validity.

### 4.2. Discussion

After further analyzing the word cloud results, some prominent elements regarding the expression of emotional status can be found. “Anxiety” is a word which is reflected in the word cloud. As is any other word on the word cloud, it is a neutral word. However, according to the overall context of the lockdown, the expectation would be of a negative relationship and a negative impact of the pandemic and the lockdown on anxiety levels and their management. If we looked at the open text individual comments, it can be observed that anxiety is often related to stress, discomfort, preoccupation, and poor management. For example, “the lack of information and the high levels of uncertainty was sufficient to provoke high levels of stress, anxiety”. Other responses mentioned that coping with high levels of demand within the home office in combination with homework caused anxiety peaks. This is aligned with numerous publications showing that students have been highly vulnerable to mental health issues during the COVID-19 pandemic and researchers have shown that perceived stress and mental health problems have increased during the pandemic [29]. In summary, an expected finding of this study is the direct impact of COVID-19 and lockdown on anxiety levels and their management among students and the university population.

Another term to highlight among the word cloud is “family”. It requires further exploration in order to understand the dimension of its impact. The comments show that there are important nuances. Some participants expressed “family” as a positive asset: “The good thing is that I still have my internship job and my family has food and health” and ”the time with my family has been excellent, being with them and feeling their care and love”. This sentiment is aligned with the literature, in which some recent studies have reported that family income stability, living with parents, and overall social support were protective factors against anxiety [29]. Other respondents noted contagion: “On the other hand, COVID-19 has caused me anxiety about the possibility of contagion in my home, keeping my distance at home in case of contagion is being very hard”. Other comments were related more to the respondents’ obligations concerning taking care of their families: “The most complicated thing has been the family conciliation” and “It is very stressful to be under four walls with all your family day and night, even more, when you are a person who likes solitude; not having an economic income to contribute to your family or home worsens your state of mind”. In conclusion, the issue of “family” is related to both negative and positive effects and our findings are aligned with those of the current literature; for example in [29] the authors have recently and surprisingly concluded that those who have spent the pandemic alone have felt less effects than those who have spent it in company. An unexpected finding of the present study was that those participants who were living with others had increased anxiety compared to those who lived alone. This finding is in contrast with the findings of previous studies, since living alone has been found to increase the risk of developing common mental disorders, such as anxiety and depression. This aspect may require further exploration.

Confinement due to government measures in order to contain the COVID-19 pandemic has had important repercussions on the lives of people in general. For university students and university staff, this situation accelerated the change that was already taking place, especially in universities, with virtual education. In Colombia, according to Forbes [30], before the pandemic only 10% of university students received virtual classes, whereas after the pandemic the Ministry of Information and Communication Technologies reports that virtuality has increased by more than 70% [31].

This study allows us to confirm the reality of students and university staff in relation to the feelings that the COVID-19 pandemic has awakened in these groups. Regardless of the type of university in which they develop their higher education training process and the tasks that are developed by the staff in the universities, there is a marked identification of negative feelings about the ways of facing the pandemic with the confinement and what this has meant for their lives.

On the other hand, sentiment analysis, in this study, is the methodology that was used to identify the positive, negative, or neutral feelings of the people from the analysis of the narratives with which the students and staff of the universities gave their accounts of how they perceive or face the COVID-19 situation. Among the words that mark their expressions is “confinement”, which marks a determining condition in people’s lives. A study conducted at the public University of San Francisco de Paula Santander in Colombia [32] shows how the students at this university have been affected in their mental health because of the confinement; since it is one of the measures that has had the greatest impact on the lives of students, affecting their personal relationships, their academic performance, and their conditions of university student life, thus generating accelerated changes in their way of life.

ASCUN, the Colombian Association of Universities, in Bulletin No.3 of August 2020 [33], projected a decrease of more than 50% in enrolments due to the pandemic. One of the major concerns was based on the lack of the technological infrastructure capacity that is required to respond to the needs of low-income students who do not have the technology and sufficient resources to turn to this new form of education, as well as the lack of employment in many families because of their socioeconomic conditions which may postpone the university studies of their children in undergraduate and graduate courses.

Another of the discussions that were raised in the ASCUN report and ratified through this study is the concern of students about the quality of the education that is received through virtual platforms, especially for those students whose training has a practical component that involves research and training developments in contact with the community, specific groups, or in laboratories [33]. Students have negative feelings about the quality of this education and feel that they will not be sufficiently prepared for their professional future, especially those students of careers that are related to the exact sciences and social sciences, whose training involves internships and practical training.

In a study that was conducted at the Universidad Francisco de Paula Santander [32], a higher incidence of depression was identified in men in the population aged between 16 and 35 years. In this present study it was not possible to differentiate feelings between men and women, nor between age groups, so it is not possible to use the presently described methodological tool in order to identify the tendency of these feelings towards positive, negative or neutral, since the analyzed expressions do not allow the identification of these characteristics.

In a study of the Pedagogical and Technological University of Colombia [34], which focused on the affectation of emotions in students and the way they face them due to uncertainty and fear of the unknown, it was shown how these circumstances have increased stress in the face of academic tasks and commitments and the way in which this negatively influenced their academic performance. On the other hand, the study also shows those feelings that are not recognized as positive or negative and that, therefore, are categorized as neutral because they express that there are particular situations of confinement which it is not possible to classify; such as those which describe being with the family, integrating into their daily activities, and the situations of other spaces that were previously controlled or autonomous for the students and are now shared with members of their family or coexistence nucleus.

We have demonstrated the feasibility of implementing NLP techniques for the sentiment analysis of the perceptions and behaviors of university staff and students using a dataset that was collected after the beginning of the COVID-19 outbreak in Colombia and Spain.

The lockdown is a government measure to contain the COVID-19 pandemic that has unevenly affected our lives. For university students and university staff, the lockdown has sped up an already underway change. Indeed, in Colombia before the pandemic, only 10% of university students received virtual classes, while after the lockdown, that number increased by more than 70% [31]. Measuring the impact of the COVID-19 pandemic on mental and physical health status may contribute to the elucidation of the risks of restrictions such as the lockdown and social distancing. Our results are aligned with publications showing that students have been highly vulnerable to mental health issues during the COVID-19 pandemic. Nevertheless, our approach is novel using AI techniques as compared to the before-mentioned studies which used other methods to determine the sentiment analysis of online survey responses.

We showed how the feelings of Colombian and Spanish students have awakened due to the COVID-19 pandemic. Unfortunately, there is a marked identification of the negative feelings about facing the pandemic with the confinement and what it means for their lives, regardless of the university that the student attended to receive their higher education or the university staff member’s duties. In addition, we identified a considerable difference in feelings between students in Spain and Colombia. It could be expected that countries with more significant development and economic infrastructure capacity, such as Spain, would have shown a higher number of positive responses compared to countries such as Colombia, which have fewer economic and infrastructure resources plus marked poverty and a social inequality gap. Paradoxically, Spain’s results showed negative feelings in more than 93% of the answers, while Colombia’s negative feelings were roughly more than 70% of the total of the analyzed answers.

The staff’s situation is very similar between the two analyzed countries: both countries showed 50% negative responses overall. However, it should be noted that Spanish university staff reported positive rather than neutral feelings in the remaining 50%, while Colombian university staff distributed their positive and neutral responses in equal proportions. The lack of noticeable differences may be explained by the fact that both country’s universities’ staff develop their tasks by using and training on similar available technologies such as Zoom and Microsoft Teams. However, there is an increasing concern about how the COVID-19 pandemic will modify education. In Colombia and other low–middle income countries, the concerns revolve around the lack of technological infrastructure capacity to respond to the needs of students and personnel, foreseeing a decrease of more than 50% in enrollment due to the COVID-19 pandemic. Another major preoccupation is how careers with a high hands-on component, such as community services for social sciences and laboratory practices for engineer careers, will be affected by virtual learning. Our methodology could be useful to identify students with negative feelings and those who are potentially more vulnerable to adapting to post-pandemic educational strategies.

All in all, understanding the impact of COVID-19 on individuals’ mental and physical health statuses can contribute to understanding the risks of severe restrictions such as lockdown and social distancing measures. Sentiment analysis has proven its validity to contribute to understanding the changes and patterns of such restrictions on overall health and wellbeing. Yet this research sheds light on the limitation of the current AI models to predict with precision further implications, due to the limitations that are explained above.

## 5. Conclusions

The proposed NLP sentiment analysis model has proven its validity to understand the changes and patterns of COVID-19 pandemic restrictions on overall health and wellbeing. In future surveys, the suggested approach using AI for sentimental analysis may be useful to uncover the impacts of the COVID-19 pandemic (or indeed any other factor) on perceptions and behaviors, reducing the manual labor that is required to apply the quality control of misclassifications. Yet, this research also sheds light on the limitation of the current AI models to predict further implications and streamline its implementation.

The methodology allows us to identify a paradox regarding the feelings that students express and shows the reality in Latin American countries such as Colombia and European countries such as Spain. The study presents a marked difference of feelings between students from Spain and Colombia; in the former group the percentage of answers that were identified with negative feelings reached a little more than 93%, leaving a small percentage between positive and neutral feelings; for the latter group, in Colombia, the negative answers were a little more than 70% of the total of the analyzed answers. This result is striking, given that it is expected that countries with greater development and economic infrastructure capacity would have a more positive response due to their conditions of adaptability to the contingent situation, whereas the reality of the findings was that Columbia, one of a number of countries which have fewer economic and infrastructure resources and a marked poverty and social inequality gap, presented a smaller proportion of negative responses, leaving between neutral and positive responses almost 30% of the students’ responses.

With respect to the staff, the situation is very similar between the two countries, both have 50% of negative responses to the situation of confinement due to the pandemic. What is striking about these responses is that the staff in Spain have positive feelings rather than neutral in the remaining 50% of responses, while the staff in Colombia share their positive and neutral responses in equal proportions. This could be explained by the fact that the Colombian staff develop their tasks in very similar conditions to the Spanish staff and that the changes that arose from the pandemic can be assimilated to the tasks that the universities develop in general. Furthermore, the use of technologies, the availability of technological tools, the possibilities for training and the training that is given to the two groups of staff are similar.

## Figures and Tables

**Figure 1 ijerph-19-05705-f001:**
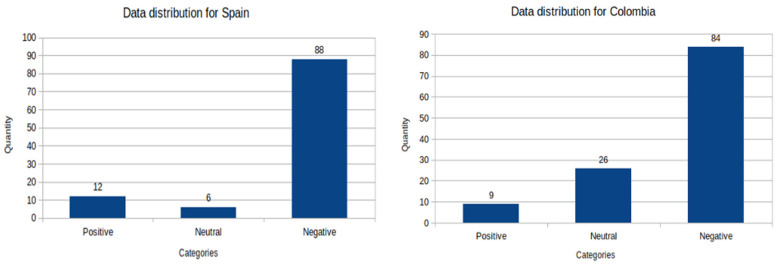
Histogram of student data distribution of countries Spain and Colombia. Source: elaborated by the authors from survey data.

**Figure 2 ijerph-19-05705-f002:**
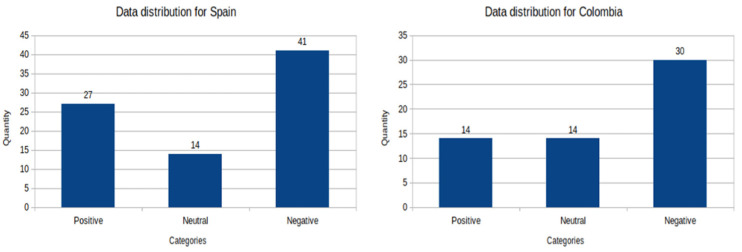
Histogram of the distribution of the data of the staff set for Spain. Source: elaborated by the authors from survey data.

**Figure 3 ijerph-19-05705-f003:**
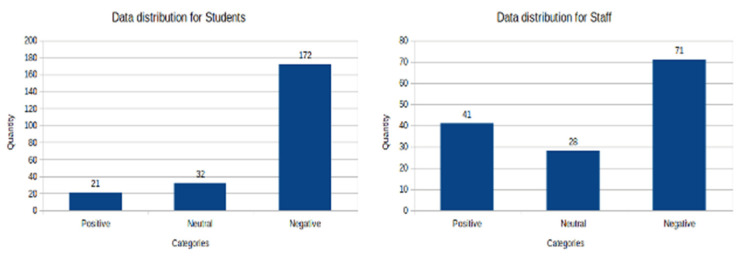
Histogram of the distribution of the data, per sentiment categories, of the students and staff datasets for two countries, Spain and Colombia. Source: elaborated by the authors from survey data.

**Figure 4 ijerph-19-05705-f004:**
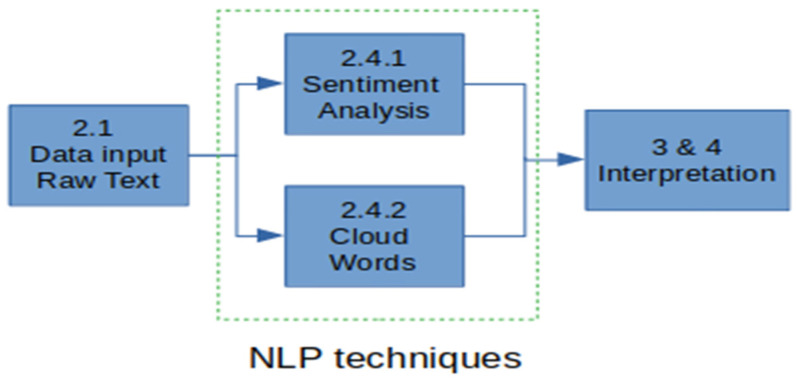
Proposed solution model based on natural language processing.

**Figure 5 ijerph-19-05705-f005:**
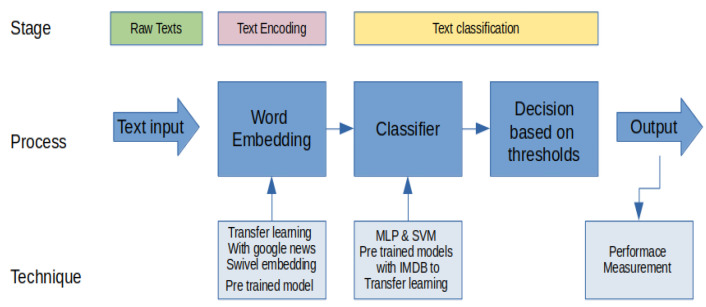
Diagram of the sentiment analysis of the input texts.

**Figure 6 ijerph-19-05705-f006:**
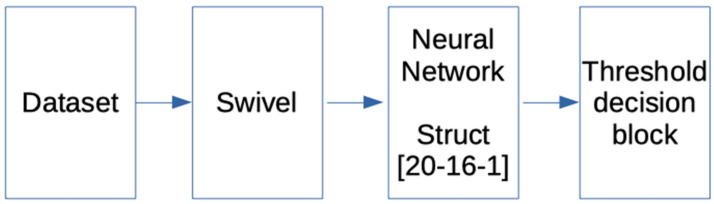
Outline of the proposal with MLP.

**Figure 7 ijerph-19-05705-f007:**
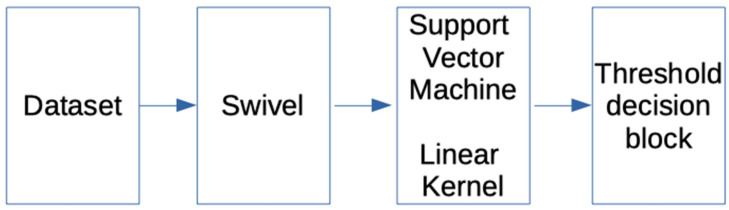
SVM proposal outline.

**Figure 8 ijerph-19-05705-f008:**
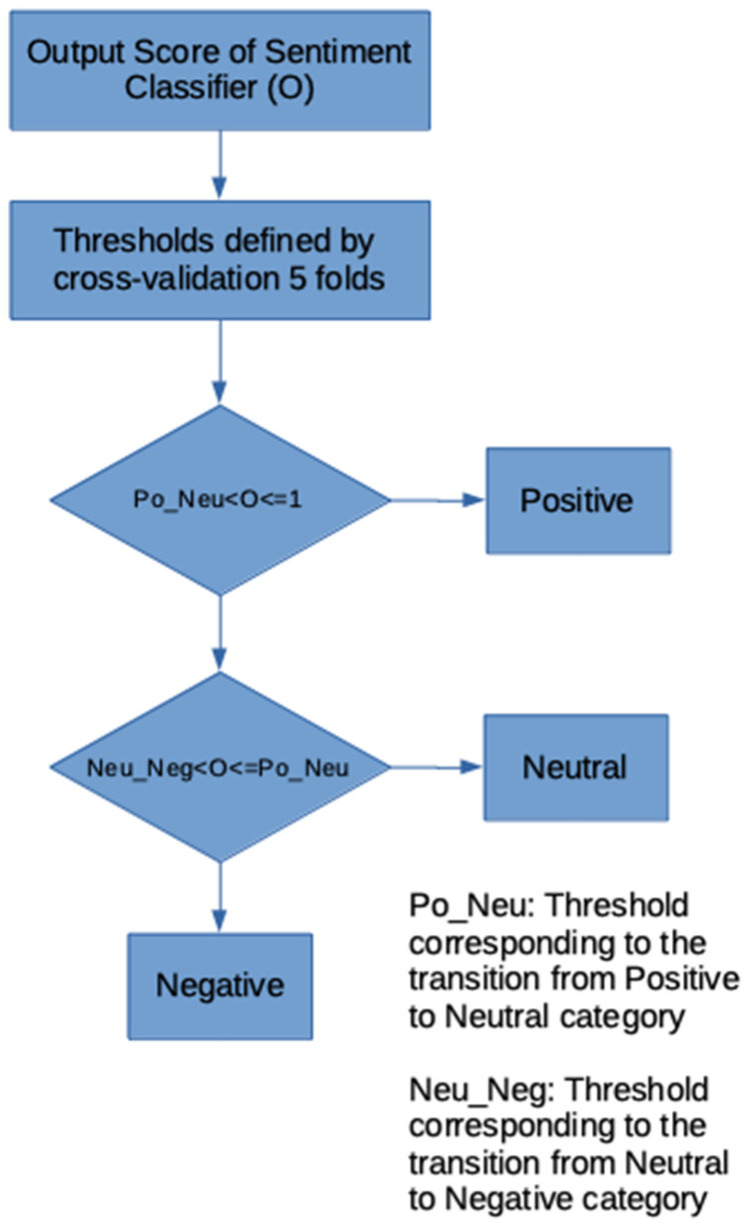
Flowchart of the decision block to determine the text sentiment.

**Figure 9 ijerph-19-05705-f009:**
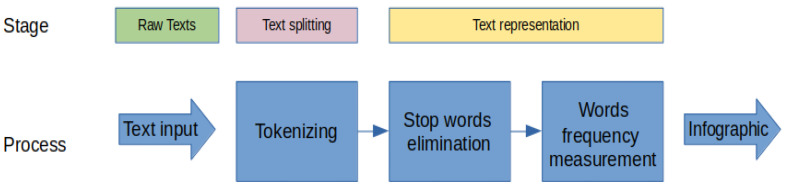
Frequency-based infographic retrieval diagram to obtain the cloud words.

**Figure 10 ijerph-19-05705-f010:**
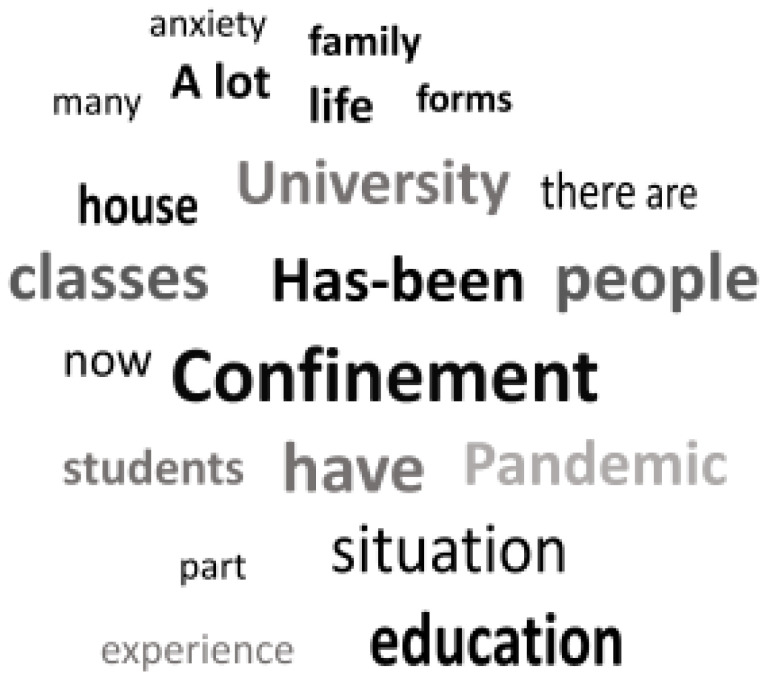
Infographic of the analyzed text corresponding to the open question of the data collection instrument.

**Table 1 ijerph-19-05705-t001:** Students and staff responses sorted by country. Source: elaborated by the authors from survey data.

Country	Students	Staff
Spain	106	82
Colombia	119	58
Subtotal	225	140
Total	365	

**Table 2 ijerph-19-05705-t002:** Average decision thresholds for each proposed model and respective dataset.

Model	Dataset	Positive Max	Mean Positive to Neutral Threshold	Mean Neutral to Negative Threshold	Negative Min
MLP	Students	1	0.49	0.22	−1
MLP	Staff	1	0.52	0.38	−1
SVM	Students	1	0.41	0.24	−1
SVM	Staff	1	0.42	0.26	−1

**Table 3 ijerph-19-05705-t003:** Metrics obtained for the present work for the different datasets and different classifiers.

Metrics	MLP Students	MLP Staff	SVM Students	SVM Staff
Weighted Accuracy	92.49%	92.59%	88.42%	82.55%
Weighted Precision	88.89%	88.57%	83.11%	72.86%
Weighted Recall	88.64%	88.47%	82.86%	71.77%
Weighted F1 Score	88.74%	88.29%	82.88%	71.75%
Accuracy	88.88%	88.57%	83.11%	72.85%

**Table 4 ijerph-19-05705-t004:** Confusion matrix obtained for the set of students with the proposed MLP model.

	Class	Real		
	Negative	165	8	2
	Neutral	1	22	6
	Positive	6	2	13
Predicted	Class	Negative	Neutral	Positive

**Table 5 ijerph-19-05705-t005:** Relevant misclassifications of the classification model (positives labeled as negatives) for the student set.

Positives Classified as Negatives
Solidarity
With the pandemic, I have come to appreciate what we have

**Table 6 ijerph-19-05705-t006:** Relevant misclassifications of the classification model (negatives labeled as positives) for the whole student body.

Negatives Classified as Positives
Most college students suffer greatly
that give a good tuition discount
The response of the organizations (Government, University...) was insufficient, with little information and a great deal of uncertainty, which is what has caused the greatest source of stress, anxiety and discomfort. In addition to a feeling of helplessness and vulnerability.
The inequality of memories and the impossibility of many families without internet to continue their studies has not been taken into account.
I am a university student. the workload has been multiplied by 3 because the professors consider that ‘‘we have more time because we are all at home’’. the accumulation of work is embarrassing and unjustifiable.
The pandemic has contributed to the fact that the millennial generation is having a very difficult time finding job stability in line with their studies.

**Table 7 ijerph-19-05705-t007:** Confusion matrix obtained for the staff ensemble with the proposed MLP model.

	Class	Real		
	Negative	70	3	5
	Neutral	0	22	4
	Positive	1	3	32
Predicted	Class	Negative	Neutral	Positive

**Class**

**Table 8 ijerph-19-05705-t008:** Relevant misclassifications of the classification model (positives categorized as negatives) for the staff.

Positives Classified as Negatives
Countries have the opportunity to learn how to improve family reconciliation, the fight against environmental pollution, teleworking and non-face-to-face or mixed modality in education, the opportunity to decrease the use of paper money and reduce monetary fraud, use national labor in jobs imported by foreign workers, improve the population rate in villages and reduce the decline in rural population, increase the hours dedicated to exercise, etc.
I have learned to better manage my time, my financial resources and my relationships with other family members. I was not working during the confinement time, which has helped me to focus on the family. I find it very difficult to reconcile work and family life if I have to work with my children at home.
I am an administration and services staff at my university. I work from home almost as before, using the family ADSL line. Working at home allows me to have sunlight, while my workstation is artificially illuminated. Now, my biggest concern is to be able to continue working from home in telecommuting mode. We are not allowed to bring our equipment home. I believe that teleworking would avoid commuting and contribute to the improvement of the environment.
The confinement has made me reflect on how fast everyday life was going: work, social life... that we are very vulnerable, that we hardly have time to dedicate to what we like or to our friends...
This experience has made me think that for some sectors, telework is a reliable and beneficial option for workers. I think it has to continue even when we come out of this pandemic. Giving the option to work from home has to be seriously contemplated not only as a preparedness measure for possible resurgences but also as a tool to improve the well-being of the workers.

**Table 9 ijerph-19-05705-t009:** Relevant misclassifications of the classification model (negatives labeled as positives) for the staff set.

Negatives Classified as Positives
It greatly reduces the quality of work, i.e., performance and concentration.

**Table 10 ijerph-19-05705-t010:** Confusion matrix obtained for the set of students with the proposed SVM model.

	Class	Real		
	Negative	160	11	3
	Neutral	8	18	9
	Positive	4	3	9
Predicted	Class	Negative	Neutral	Positive

**Table 11 ijerph-19-05705-t011:** Confusion matrix obtained for the set of staff with the proposed SVM model.

	Class	Real		
	Negative	66	5	11
	Neutral	3	15	9
	Positive	2	8	21
Predicted	Class	Negative	Neutral	Positive

## Data Availability

Data were obtained from Osipenko [35].

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
