# Peer review of "Analysis of the Effects of Lockdown on Staff and Students at Universities in Spain and Colombia Using Natural Language Processing Techniques"

_ijerph, 2022, doi:10.3390/ijerph19095705_

Round 1
Reviewer 1 Report
The article as a whole is well-established, and the subject is interesting.
The consequences of lockdown on the academic community (faculty and students) are demonstrated through the use of text mining techniques based on artificial intelligence algorithms. Additionally, the present study benefits from the merging of two distinct nations that have significant commonalities.
Numerous published papers on the impacts of lockdown on individuals make use of materials and methods like as electronic surveys to illustrate the investigated population's perspectives in a descriptive manner. In this case, the use of artificial intelligence models provides a novel viewpoint on the subject. Additionally, this research benefits from the merger of two distinct countries (multicountry perspective).
The paper is well-established and written in an easy and understandable manner.
The study's results are consistent with the facts and arguments offered and are well documented. The concluding part addresses and thoroughly explains the criteria that have been considered as key issues throughout the pandemic: family, anxiety, house, and life.
Author Response
"Please see the attachment."

Reviewer 2 Report
A brief summary
The paper presents the use of sentiment analysis to diagnose the mood of the Spanish and Colombian academic community after COVID-19 pandemic.
The aim of this paper is to propose tool for diagnose the state of mind of students and university staff after lockdown related to the COVID-19 pandemic.
The main contribution are some findings about life priorities among students and academic staff during lockdown.
Broad comments
1) It is very interesting idea to analyse academic sentiment during Covid-19 pandemic with neural networks methods. The general goal here is to classify messages. The question is "what for?". It must be expalined.
2) In my opinion, the use of automatic translation for sentiment analysis is wrong. Whether it is due to the accuracy of the translation or due to cultural differences, this approach may produce erroneous results. Sentiment analysis should be done on the words in which the message was expressed. The presented paper will be a contribution to the discussion on this view.
3) The numerous differences in the presented data on the number of responses raise suspicions as to the reliability of the data presented
Specific comments:
l. 91 - actually, there are two mentioned classes: first with source in social networks and second with source in texts from interviews; the Twitter sources belongs to the first one
l. 149 - mentioned 12 countries does not correspond to presented on Figure 1 and Table 1, two countries
l. 187 - mentioned 13 Positive answers does not correspond to presented on Figure 2 twelve Positive
l. 198 - mentioned number of answers does not correspond to presented on Figure 4
l. 219 - mentioned 73 Negative answers does not correspond to presented on Figure 7 71 Negative
l. 327 - symbols used in Figure 12 should be previously explained, to be able to assess the coherence of the definition of the positive and negative intervals
l. 360 - the number of records didn't correspond to presented previously
l. 374 - as negative message has value -1, Mean Neutral-Negative threshold should be below zero.
l. 496 - Solomou et al it is perhaps position [29]
l. 604 - perhaps there should be AI in place of IA
Author Response
"Please see the attachment."

Reviewer 3 Report
Summary:
This paper proposes to analyze the impact of COVID-19 on the university community jointly on staff and students of two countries using Natural Language Processing techniques. The authors find that the most often related words are family, anxiety, house and life when discussing personal reflections. The authors also find that staff have a slightly less negative perception of the consequences of COVID in their daily life. An online survey was conducted to collect responses from 225 students and 140 staff members.
Strengths:
The authors claim to be the first of this kind to conduct sentiment analysis using NLP techniques in online public opinion surveys.
Responses from 225 students and 140 staff members of two counties are collected during the experiments.
Weaknesses:
This submission seems like an on-going project, and more work needs to be done. For example,
- To classify the three categories (positive, neutral and negative), a manually selected threshold is needed, which is not a general practice in machine learning models (fig. 10, fig.11).
- Each open response was labeled as positive, negative or neutral by two independent researchers, but what’s the agreement between these two researchers? What if one sentence contains both positive and negative attitudes towards COVID? What is the relationship between the labels and the other 67 closed-end questions?
- The data distribution is unbalanced (positive, neutral and negative), please report F1 score as well for performance evaluation.
- The task validated in this submission is a very naive task, please add more tasks to fully utilize the collected dataset, such as: can we classify student vs staff based on the 68 reponses? Which of the 67 closed-end questions is closely related to the open-end questions? Etc
- Figures need to be revised. For example, Fig. 2 and Fig.3 can be combined into a single figure, the same for Fig.4 and Fig. 5.
Based on the weaknesses as mentioned above, I give my score for this submission.
Author Response
"Please see the attachment."

Reviewer 4 Report
Thank you for the opportunity to review the article “Analysis of the effects of lockdown on staff and students at universities using Natural Language Processing Techniques: case study of Spain and Colombia”. The paper addresses an interesting and well researched theme in the recent period since the pandemic with this first observation that this study is the first attempt to analyze the lockdown effect using Natural Language Processing Techniques, particularly sentiment analysis methods applied at large scale.
This study represents a solid effort in the field approached. It is constructed in a mature manner, following the publication standards of the journal.
Also, the study is written in an adequate manner, with a generous review of the literature and robust research design. The results are presented clearly and coherently.
As the authors say, one conclusion of this paper assumes that “the most often related words were family, anxiety, house and life. On another front, it has also been shown that staff has a slightly less negative perception of the consequences of COVID in their daily life.”
The strong points of this article are the ability to propose a comprehensive research design with sufficient number of respondents (365) to present the data for the present research and the bi-directional perspective of the study, taking account of the students and the staff opinions.
The weak points are represented by the lack of a section in the article discussing future directions, presenting Limitations and recommendations for future research and the underrepresented reference list regarding the studied topic.
Moreover, there are some point-by-point observations that should be addressed in this revision.
- Line 70, (Imanol, O.). – please use the journal style for citation
- Line 79 "The current situation of education in Spain on the 79 advice of the pandemic," this idea should be cited and mention the page number
- The whole text should be verified for English language again (e.g., For the country of Spain, For the Colombian country, etc.)
- Use one coherent version for `COVID-19`, as in text there are found several versions (COVID 19, COVID, Covid-19, etc)
- Rethink the title of the article when discussing about `case study`. First of all there are two cases, Spain and Colombia, second, the authors are presenting a secondary analysis of the dataset from the research using the responses of an open-ended question: “The data were obtained through a web-based comprehensive questionnaire composed of 68 questions, 67 closed-ended questions that cover different aspects: socio-demographic, geographic, and the impact of the COVID pandemic in their daily lives (studies, work, families, social life, habits…). The last question was an open-end question related to personal reflections: “Share any other thoughts / experiences about your life in lockdown” and set the basis for the analysis of this paper.” and third, the authors are not discussing the case study method (as considered by Robert Yin, https://us.sagepub.com/en-us/nam/case-study-research-and-applications/book250150) in this article.
- Figure 1 `Map highlighting countries of origin to the Spanish survey` makes no sense just to present two countries from the geographical point of view.
- All the `2.3.1. Histograms of labeled text by sentiment categories` section should be reorganized because all the six graphics used are mentioning a very simplistic descriptive analysis that could be synthetized in a couple of paragraphs.
- Use objective text, not `Our sentiment analysis` (Line 607)
- Use the journal style for the references.
Author Response
"Please see the attachment."

Round 2
Reviewer 4 Report
Thank you for the opportunity to review the paper “Analysis of the effects of lockdown on staff and students at universities using Natural Language Processing Techniques: case study of Spain and Colombia”.
The authors responded/ explained with reasonable arguments and corrected all the remarks and observations highlighted in the second version of the manuscript provided and the results suggest a more consistent and logical text with a clearer reference list.
To sum it up, the authors developed a more in-depths theoretical presentation about the subject, integrating all the suggested aspects and the conclusions of the review are also modified and the reference list is updated.
I consider that the paper is publishable after a final check from the authors.
This manuscript is a resubmission of an earlier submission. The following is a list of the peer review reports and author responses from that submission.
Round 1
Reviewer 1 Report
Paper presents the results of NLP analysis of the text of students and teachers messages in Spain and Columbia regarding Covid 19.
Paper is interesting with good references materials.
However, the formatting of the paper is not clear. That is why I have several suggestions:
1) Page 3, text after 2. Materials and methods - text should be removed (this is part of instruction)
2) Please add into Introduction the main contribution of the paper
3) What is the purpose of Fig. 1? The important information is in Table 1. I think Fig 1 should be removed
4) Is the survey presented in open-source form? How can we prove the results?
5) Figures 2-7 should be given in one table.
6) The enumeration of subsections is missed. For example, the authors refer to section 2.4.1.1. (page 9), but below this section is presented without numbers.
7) MLP: what is about hyperparameters of a neural network? What are the motivation to choose 20 input neurons and 16 hidden neurons?
8) The same for SVM: please add parameters and hyperparameters
Reviewer 2 Report
The authors perform a linguistic analysis to study the impact of Covid-19 in the life of university students and staff.
The work must be improved in each of its aspects. The presentation is not clear, the authors do not describe related works and approaches, the description of the methods lacks a lot of important information, and I suggest the authors to use a language proofreading service. With so much information missing, it is impossible to evaluate the scientific merit and soundness of this work.
Main comments:
1) The authors claim this is the first study of this type, but do not provide any description of related works. Despite their claim, there are plenty of works on this topic. Just to cite some:
- Sentiment analysis and its applications in fighting COVID-19 and infectious diseases: A systematic review - Alamoodi et al
- Sentiment analysis of nationwide lockdown due to COVID 19 outbreak: Evidence from India - Barkur et al
- Twitter Sentiment Analysis during COVID-19 Outbreak - Dubey
but many more can be easily found with a rapid search.
The authors must describe related works in an appropriate section, and specify similarities and differences with their work.
2) The whole work is based on the analysis of a single question out of 68. I think taking at least some of the additional questions into account would be insightful.
3) Regarding the annotation process, the authors must specify more details: Each response was labelled by both the annotators or only by one? The annotation guidelines were discussed before or during the annotation process? Has the inter-annotator agreement been measured? How have the annotators handled sarcasm and/or irony?
4) Regarding word embeddings. Why have the authors chosen GloVe rather than more modern techniques such as Elmo, Bert, or Sentence-Bert? The choice of GloVe can be okay, but must be motivated. How have you handled out of vocabulary words?
5) How are the 5 folds created? Randomly or by enforcing constraints on the distribution of the labels?
6) In an unbalanced dataset like this, accuracy is not very informative: a very stupid classifier that always predicts the majority class may obtain a good accuracy for example. It would be more appropriate to use F1 macro metric. Also, I suggest evaluating simple baselines like random and majority class classifiers to help understand whether the task is difficult or trivial.
7) The purpose of this work is quite not clear. The authors say that they want to analyze the impact of Covid-19, but it seems that they have not designed a proper analysis framework and established research questions. Instead, they seem to have taken the questionnaire answers and run some simple experiments on which they have built ad-hoc conclusions. The authors draw a lot of complex conclusions based on the results they have obtained with word-clouds, tools that simply present the most frequent words. If the authors want to back their claims, they must perform a more detailed analysis.
8) The fact that an automatic translation tool is used MUST be mentioned in the method! Details about which tool is used must be included. Moreover, why using a translation tool and not instead use Spanish word embeddings or multilingual word embeddings?
Minor comments
1) In section 2 (Materials and Methods), the authors have forgotten 3 entire paragraphs from the template!!!
2) Figure 1 is completely pointless
3) The initial paragraph of section 2.3.1 is impossible to understand
4) The "histograms" sections seems unnecessary, a simple table with the numbers would be enough to report the dataset composition
5) What are the "raters" mentioned on page 10?
6) Figure 12 seem a verbose explanation of a rather simple concept
7) What is the meaning of column Positive and Negative in Table 2?
8) Conventionally, in the confusion matrices, the real classes are in the rows, the predictions in the columns.